# Therapeutic Targeting of P53: A Comparative Analysis of APR-246 and COTI-2 in Human Tumor Primary Culture 3-D Explants

**DOI:** 10.3390/genes14030747

**Published:** 2023-03-19

**Authors:** Adam J. Nagourney, Joshua B. Gipoor, Steven S. Evans, Paulo D’Amora, Max S. Duesberg, Paula J. Bernard, Federico Francisco, Robert A. Nagourney

**Affiliations:** 1Nagourney Cancer Institute, 750 E. 29th Street, Long Beach, CA 90806, USA; 2Molecular Gynecology Laboratory, Gynecology Department, College of Medicine of the Federal University of São Paulo (EPM-UNIFESP), Rua Pedro de Toledo, São Paulo 04039-032, Brazil; 3Department of Obstetrics and Gynecology, University of California Irvine (UCI), 101 The City Drive South, Orange, CA 92868, USA

**Keywords:** TP53, APR-246, COTI-2, anti-cancer drug

## Abstract

**Background:** *TP53* is the most commonly mutated gene in human cancer with loss of function mutations largely concentrated in “hotspots” affecting DNA binding. APR-246 and COTI-2 are small molecules under investigation in P53 mutated cancers. APR binds to P53 cysteine residues, altering conformation, while COTI-2 showed activity in P53 mutant tumors by a computational platform. We compared APR-246 and COTI-2 activity in human tumor explants from 247 surgical specimens. **Methods:** Ex vivo analyses of programmed cell death measured drug-induced cell death by delayed-loss-of-membrane integrity and ATP content. The LC50s were compared by Z-Score. Synergy was conducted by the method of Chou and Talalay, and correlations were performed by Pearson moment. **Results:** APR-246 and COTI-2 activity favored hematologic neoplasms, but solid tumor activity varied by diagnosis. COTI-2 and APR-246 activity did not correlate (R = 0.1028) (NS). COTI-2 activity correlated with nitrogen mustard, cisplatin and gemcitabine, doxorubicin and selumetinib, with a trend for APR-246 with doxorubicin. For ovarian cancer, COTI-2 showed synergy with cisplatin at 25%. **Conclusions:** COTI-2 and APR-246 activity differ by diagnosis. A lack of correlation supports distinct modes of action. Cisplatin synergy is consistent with P53’s role in DNA damage. Different mechanisms of action may underlie disease specificity and offer better disease targeting.

## 1. Introduction

The *TP53* tumor suppressor is the most commonly mutated gene in human cancer [1]. The P53 protein functions by detecting and responding to oncogenic stress. Cells experiencing DNA damage, metabolic dysfunction, hypoxia, oncogene expression or replicative stress activate P53 to initiate cellular responses that include but are not limited to DNA repair, cell cycle arrest, cell senescence and apoptosis [2].

The P53 protein functions primarily as a transcription factor. The functionally active homo-tetramer binds to specific response elements within the genome driving a multitude of responses. Minor DNA damage elicits cycle arrest and activates DNA repair while more severe DNA damage leads to cellular senescence or apoptosis [3].

P53 induces apoptosis by upregulating *Puma*, *Noxa*, *Bax*, *Apaf1*, *Fas*, *Tnfrsf10B/DR5*, miR34, *TP53AIP1*, *Pidd*, *Pig3*, *Zmat3* and *Siva*. *Puma*, *Nova*, *Bax* and *Apaf1* are part of the intrinsic pathway, while *Fas* and *Tnfrsf10B/DR5* are mediated by the extrinsic pathway. The microRNA miR34 that sensitizes cells to apoptosis is also under P53 control [4]. 

P53 regulates the cell cycle by inducing G1/S arrest, providing time for the detection and repair of DNA damage. P53 also induces G2/M arrest, regulating cell proliferation and responding to metabolic deregulation. 

P53 induces senescence through *Cdkn1a*, *Pml*, *Pail* and *E2f7*, preventing the progression of premalignant lesions to a state of malignancy [5].

An important function of *TP53* is to regulate metabolism by stimulating oxidative phosphorylation and inhibiting glycolysis [6]. P53 downregulates the expression of GLUT1, GLUT3 and GLUT4 transporters, thereby decreasing glucose uptake as a substrate for glycolysis. P53 also activates *TIGAR*, further inhibiting glycolysis and increasing the production of NADPH [7]. 

P53 also functions as a protein modulator. During ischemia-related oxidative stress, P53 accumulates in the mitochondrial matrix through its interaction with cyclophilin D, leading to the opening of the permeability transition pore and, ultimately, necrosis [8,9]. P53 also binds to glucose-6-phosphate dehydrogenase (G6PD), the rate-limiting enzyme of the pentose phosphate shunt and decreases flux [10].

P53 functions as a tetramer comprised of four identical monomers. The monomer structures within the tetramerization domain consist of a β-sheets linked to an α helix at Glycine 334. These monomers then dimerize through interactions with their α-helices and β-sheets giving P53 its characteristic “dimer of a dimer” structure. 

Within the primary structure, different domains contribute to P53’s pro-apoptotic function. The first 100 amino acids are categorized as the transactivation domain, with a proline-rich region, residues 64 to 92, that is necessary for transactivation. The second domain, residues 102 to 295, is the DNA-binding domain. Amino acid residues 295 to 323 are the nuclear localization domain, while amino acid residues 325 to 356 are the tetramerization domain. The remaining amino acids are the C-terminal domain [11,12].

The extremely broad array of anti-oncogenic functions of P53 has made this pathway an attractive target for cancer therapy. Among the approaches has been the use of a recombinant adenovirus (Gendicine) [13] to deliver functional TP53 expression in transformed cells. This was approved for use in combination with radiation by the Chinese Food and Drug Administration in 2003. An alternative approach has been to target G2/M arrest with WEE-1 inhibitors that selectively target P53 mutants that have lost P53-mediated G1/S arrest, resulting in a mitotic catastrophe [14]. 

As P53 has broad and diverse functions, its role in different diseases varies [15]. Working in concert with KRAS in pancreatic cancer, P53 provides potent survival signals and a high degree of drug resistance [16]. Similarly, *TP53* loss (deletion 17) in chronic lymphocytic leukemia is among the worst prognostic findings [17], yet high-grade serous ovarian cancer, the vast majority of which carry P53 missense mutations [18], remain highly sensitive to platinum-based chemotherapy. Moreover, the gain of function mutations in lung cancer has been suggested as a druggable vulnerability [19]. In light of the complexity of P53 function, we applied a cellular response platform to examine the effect of P53 active agents. In recent years, two novel small molecules have entered clinical trials: APR-246 and COTI-2.

Eprenetapopt (APR-246) is a novel, first-in-class, small molecule that selectively induces apoptosis in *TP53*-mutant cancer cells. Eprenetapopt is a prodrug that, under physiological conditions, is spontaneously converted to methylene quinuclidinone, a Michael acceptor that covalently binds to cysteine residues in mutant P53, leading to thermodynamic stabilization of the P53 protein and shifting the equilibrium toward a functional conformation [20,21,22]. In addition, eprenetapopt increases oxidative stress by the depletion of glutathione and inhibition of thioredoxin reductase, leading to the accumulation of reactive oxygen species and further promoting tumor cell death [23,24,25,26,27]. Eprenetapopt has synergistic cytotoxicity when combined with azacitidine in *TP53*-mutant MDS and AML cell lines, primary patient specimens and in vivo models [28]. In a phase I study including patients with AML, eprenetapopt monotherapy demonstrated clinical activity with corresponding activation of P53-dependent pathways [29,30]. Recently, the US Food and Drug Administration granted a breakthrough therapy designation for the treatment of patients with *TP53*-mutant MDS with the combination of eprenetapopt and azacitidine. Recent studies have shown safety and efficacy findings from the phase Ib/II trial, which evaluated a combined treatment with eprenetapopt and azacitidine in patients with *TP53*-mutant myeloid malignancies [NCT03745716].

COTI-2 is a third generation thiosemicarbazone compound shown to exhibit anti-cancer activity against malignant cell lines of diverse tissue of origin. The compound was discovered using a proprietary computational platform known as CHEMSAS^®^, which uses a unique combination of traditional and modern pharmacology principles, statistical modelling, medicinal chemistry and machine-learning technologies to discover and optimize novel compounds for the treatment of cancer [31,32]. When compared to multiple current treatments (chemotherapy or target therapies) for specific cancer types, COTI-2 was reported to be more active in the preclinical models tested. Furthermore, in the animal models investigated, COTI-2 was found to be safe and well tolerated [32]. Recently published data suggested that COTI-2 acts, at least in part, via the reactivation of mutant P53 to a wild-type form. In particular, COTI-2 was shown to normalize a wild-type *TP53* target gene to a *TP53*-mutant. However, COTI-2 has also been reported to act independently of P53, i.e., by inhibition of the *mTOR* pathway and activation of AMPK [33].

Currently, COTI-2 is undergoing evaluation for the treatment of recurrent gynecological and other cancers in a phase I clinical trial [NCT02433626].

To examine the activity of APR-246 and COTI-2, we applied the ex vivo analysis of programmed cell death (EVA/PCD) to 247 human tumor primary culture specimens obtained from surgical procedures, cytologically positive fluids and peripheral blood that were submitted to our CLIA-licensed laboratory. 

The EVA/PCD platform has been shown to correlate with the response, time to progression and survival in human cancer [34,35]. The results provide a clinically validated laboratory model to explore the therapeutic potential of APR-246 and COTI-2 in cancer patients.

## 2. Materials and Methods

The ex vivo analysis of programmed cell death methodology has previously been described [36,37,38]. Briefly, after mechanical and enzymatic disaggregation, micro-spheroids were isolated by precise density centrifugation and exposed to APR-246, COTI-2 and other agents for 72–96 h. Drug-induced cell death was measured by delayed-loss-of-membrane integrity and ATP content (luciferase). The dose–response curves were interpolated to the LC50 values. Synergy analysis used the method of Chou and Talalay [39]. The correlation coefficients used Pearson moment.

For 49 APR-246 tested and 86 COTI-2 tested samples, genomic data was available as provided by commercial next-generation DNA sequencing (NGS) laboratories, including Foundation One, CARIS, Neo Genomics, Guardant 360, Memorial Sloan-Kettering and MD Anderson, among others. *TP53* mutations were subdivided by the type and location of the mutation.

### 2.1. Tissue Procurement and Tumor Cell Preparation

Patients undergoing surgical procedures provided sterile tumor specimens for analysis. All the patients provided written consent to allow the use of tumor biopsy specimens. At the time of surgical exploration, the tumor tissues removed from the patients were processed sterilely and placed in sterile RPMI 1640 media containing 2 mm l-glutamine, 15% fetal calf serum, 100 IU mL^−1^ penicillin and 100 μg mL^−1^ streptomycin (modified RPMI 1640) and were submitted directly to our laboratory. The samples obtained after hours were maintained at 4 °C until processed (always <24 h). 

The tumor specimens were mechanically disaggregated by sterile mincing with scalpels. The specimens were then incubated for 2 h in 0.8% collagenase IV and 0.002% DNaseI. The cells were isolated by density centrifugation over Ficoll–Hypaque, washed and then resuspended in modified RPMI 1640. The cell suspensions were adjusted to 1 × 10^6^ cells mL^−1^ and distributed into 96-well polypropylene culture plates (90 μL well^−1^). Serial dilutions of the drugs and drug combinations in 10 μL volumes were added, and the plates were incubated at 37 °C in 6% CO_2_ in a sterile, humidified environment for 72 h, as described in the next section.

### 2.2. Drug Exposure and Measurement of Cell Viability

The dose-dependent cytotoxicity of the drugs was investigated using a five-point dose–response curve. Serial dilutions of the drugs were prepared in 0.15 m NaCl, and 10 μL of a drug solution was added to each well. The cells were incubated for 72 h with saline vehicle control (0.15 m NaCl) or in the presence of APR-246 (40.2 to 2.5 μMol), COTI-2 (317 to 19 μMol), cisplatin (22 to 1.4 μMol), doxorubicin (2.2 to 0.14 μMol), gemcitabine (483 to 29 μMol), nitrogen mustard (28 to 1.8 μMol), paclitaxel (35 to 2.2 μMol), everolimus (20.9 to 1.3 μMol), selumetinib (109 to 6.8 μMol), cisplatin and gemcitabine at a fixed ratio (22 to 484 μMol) or cisplatin and COTI-2 at a fixed ratio (22 to 6.8 μMol).

### 2.3. Delayed-Loss-of-Membrane Integrity

After 72 h, 10 μL of phosphate-buffered saline containing 0.5% nigrosin-B and 1% Fast Green and 37,500 glutaraldehyde-fixed avian red blood cells (internal standard) were added to each well, and the samples were gently agitated to facilitate mixing. After 10 min, the samples were aspirated and then centrifuged onto a glass slide using a modified cytospin cassette. The air-dried samples were counterstained with modified haematoxylin and eosin. The tumor cell viability was determined as the ratio of living tumor cells over simultaneously counted avian red blood cells. The cell survival of the drug-treated samples was expressed as a percentage of the saline control values.

### 2.4. ATP Content

The ATP concentration is measured using a luciferin reagent that is converted into oxyluciferin by luciferase, with the number of viable cells in the culture shown to correlate with the luminescence over 3 orders of magnitude (ProMega CellTier-Glo Assay specifications, Promega Corporation, Madison, WI, USA). The cell viability is measured after 72 h by introducing 100 μL of CellTiterGlo solution to each well in the 96-well plate. The plates are shaken for 2 min. The cell suspension is then incubated for 10 min. The cell suspension is then shaken for an additional 2 min. The 96-well plate is read on a Fluoroskan Ascent FL instrument, according to the manufacturer’s instructions. 

The 50% lethal concentration (LC50) values were calculated using a least-squares line of best-fit generated from the five-point concentration curve, with the LC50 values interpolated from the curve. The synergy was determined using the median effect technique [40]. In brief, dose–response curves and LC50 values were generated for each agent, as a single agent and in combination with other agents. A comparison of each agent dose–response curve for a given tumor type allowed the determination of the activity and, where applicable, the synergy, The term synergy only applied to the tumor samples with 100% of the points on the combination dose–response curves falling above the line of additivity.

The overall mean LC50 (LC50T) and s.e.m. for all the tumor types (s.e.m. T) within our laboratory database were utilized to calculate the Z-scores by the following formula:(LC50sample − LC50T)/s.e.m. T

The use of inferential statistics is predicated upon the central limit theorem, a fundamental sampling theorem that states that the distribution of a sample population is unimportant so long as the samples are drawn randomly in sufficient numbers from a parent population whose distribution need not have a normal (Gaussian) distribution [40].

By normalizing the LC50 values for each single agent and/or combination around the mean LC50 values, the samples found to be more resistant than average fell to the right of the mean, whereas those that were more sensitive than average fell to the left of the mean. This Z-score transformation is routinely used in the US by the National Cancer Institute for studies comparing microarray data and other analyses [41]. It has been incorporated into the COMPARE statistical program and the latest version of the National Cancer Institute’s public access MAExplorer bioinformatics tool [42].

### 2.5. Statistical Methods

The LC50 means for each agent were compared across the tumor types, with the criterion for statistical significance set at *p* ≤ 0.05. For comparisons of in vitro activity or combinations, the LC50 values were normalized to Z-scores (with the mean set to 0 and variance from the mean in s.e.m. units of 1). This permitted direct comparisons of the sensitivity versus the resistance of single agents and drug combinations across tumor types. Using these normalized LC50 values, graphic representations were generated, depicting on the left side of the mean those samples that were more sensitive than average to drug combinations (negative range) and on the right side of the mean those that were more resistant than average (positive range). Pearson product moment correlations were used to provide R values, which were examined by a two-tailed Student *t*-test for significance <0.05. Bonferroni corrections were applied by dividing the 0.05 value by the number of separate correlations conducted.

## 3. Results

A comparison of disease-specific activity for APR-246 and COTI-2 was conducted by (SEM) Z-Score analyses (Figure 1 and Figure 2). As can be seen, the activity favored hematologic neoplasms over solid tumors for both compounds. By disease, solid tumor activity differed for the two compounds. 

To measure the impact of the TP53 mutation type, amorphic (little or no functional P53 protein) versus hypomorphic (altered P53 protein structure), we conducted a (SEM) Z-Score analysis for COTI-2 and AZD 246 that compared the drug activity with the NGS-measured mutation type (Figure 3). We provide the specific mutations identified in *TP53* measured by next-generation DNA sequencing (NGS) (Appendix A).

Representative dose–response curves using serial dilutions of APR-246 and COTI-2 to measure drug-induced cytotoxicity are provided (Figure 4a,b). 

To interrogate the mechanisms of action for COTI-2 and APR-246, we examined correlations between the compounds and other classes of drugs by Pearson moment R value. As can be seen in Table 1, APR-246 showed a trend toward association with doxorubicin, which did not persist with Bonferroni correction, while COTI-2 activity correlated with AZD 6244 (MEK/ERK), cisplatin + gemcitabine, doxorubicin and nitrogen mustard, all of which remained significant after Bonferroni correction.

A final analysis applied the method of Chou and Talalay [40] to examine the synergy between COTI-2 and cisplatin in ovarian cancer specimens, the subject of a clinical trial. Synergy was identified in 25% of the specimens, while antagonism was found in 18%.

## 4. Discussion

We undertook this analysis to compare and contrast the activity of these two novel compounds and explore the disease specificity, possible mechanisms of action, combinatorial potential and impact of specific *TP53* mutations on efficacy using a previously validated 3-D human tumor explant platform.

Our findings reveal that COTI-2 and APR-246 are cytotoxic to human tumor 3-D explants. Though structurally and mechanistically different, both agents have been developed to target tumors with P53 mutations. When we directly compared the activity for these two agents by Pearson moment, the R value of 0.1 clearly supports distinct mechanisms of action.

The activity profiles for both agents favor hematologic neoplasms. Many cytotoxic drugs are more active in blood-borne tumors compared with solid tumors. As only 20% of hematologic tumors carry P53 mutations, yet the activity profiles favor hematologics over solid tumors, even ovarian cancers, where up to 80% of patients carry P53 mutations, the activities for both compounds appear to extend beyond their direct impact upon P53. For APR-246, this may reflect its effect on the redox state via thioredoxin reductase, while COTI-2 appears to have many effects downstream of P53.

An examination of the impact of the *TP53* mutation type upon drug activity supports APR-246’s putative mode of action as one that alters the conformation of the P53 protein, in as much as only hypomorphic mutations appeared to be sensitive to this compound with virtually no activity for the amorphic subtypes. In contrast, COTI-2 activity was found in both the amorphic and hypomorphic subtypes, suggesting a mode of action that functions downstream of the P53 protein.

Exploratory analyses enabled us to examine the synergy for cisplatin combined with COTI-2 in ovarian cancer specimens. This is the subject of a clinical trial examining COTI-2 alone and in combination with cisplatin in gynecologic malignancies (NCT02433626). Based on the putative mechanisms of action of COTI-2 in DNA damage repair, the 25% synergy level was somewhat lower than we had expected.

An additional finding was a high correlation between CDDP and COTI-2 (R = 0.97, *p* = 0.02) in a small subset of specimens with amorphic P53 mutations. This will be further examined in additional samples as specimen availability allows.

The limitations of the study include an incomplete database of NGS-measured mutational status, a lack of APR-246 combination studies and the heterogenous makeup of the tumor samples studied. As clinical studies of APR-246 were largely focused on hematologic tumors, and those for COTI-2 upon gynecologic tumors, these cancer types were overrepresented in our database. We continue to add different tumor types to our studies to further characterize other types of cancers for their response to these compounds. An additional limitation was the preponderance of DNA mutations in the DNA-binding domains, which limited our capacity to examine the impact mutations in the dimerization and tetramerization domains that may be more germane to APR-246 activity.

The study was exploratory by design, as the exact mechanisms of action, combinatorial potential and disease specificity for COTI-2 and APR-246 are still under active investigation. Nonetheless, the results suggest disease targets and may point toward the more rational use of these drugs in the future based on the tumor type and *TP53* mutational subtype.

## 5. Conclusions

Breakthroughs in genomics and transcriptomics have greatly expanded our understanding of the genetic origins of human cancer. Of the nearly one thousand cancer-related genes identified to date, a large number function as tumor suppressors, with *TP53* being the most common aberrancy identified. In this context, P53 has become the target of developmental therapists seeking actionable mutations, yet the complexity, redundancy and promiscuity of downstream signals for this gene product have often frustrated attempts to develop effective drugs.

We applied a phenotypic approach to measure the impact of two novel small molecules designed to target P53, APR-246 and COTI-2, to measure cellular viability following 72 h of continuous exposure as a surrogate for clinical response. The results confirm cytotoxic activity against human tumors and suggest disease specificity, while correlative studies suggest possible modes of action.

As the ex vivo analysis of programmed cell death (EVA/PCD™) platform has been shown to correlate with the response, time to progression and survival, the insights gained from this study may allow for the better selection of treatment candidates both by disease type and individual patient profile and can facilitate the development of effective drug combinations based upon the measurement of potential therapeutic synergy. Validated primary culture techniques have the capacity to measure all the operative cellular mechanisms of drug response and resistance, acting in concert to provide clinically relevant signals for patient therapy and accelerate drug development.

## Figures and Tables

**Figure 1 genes-14-00747-f001:**
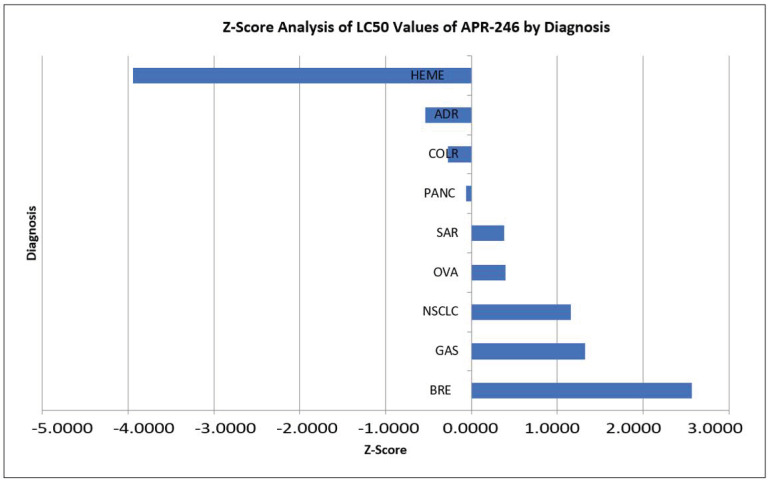
Lethal concentration (LC50) values were compared using Z-scores calculated by the formula Z = X − U/S, where X = Mean value, U = Sample value and S = Standard deviation. Results falling to the left of mean are more sensitive. Results falling to the right of mean are more resistant: Heme = Hematologic tumors; Adr = Adrenocortical; Colr = Colorectal; Panc = Pancreatic; Sar = Sarcoma; Ova = Ovarian; NSCLC = Non-small cell lung cancer; Gas = Gastric; Bre = Breast.

**Figure 2 genes-14-00747-f002:**
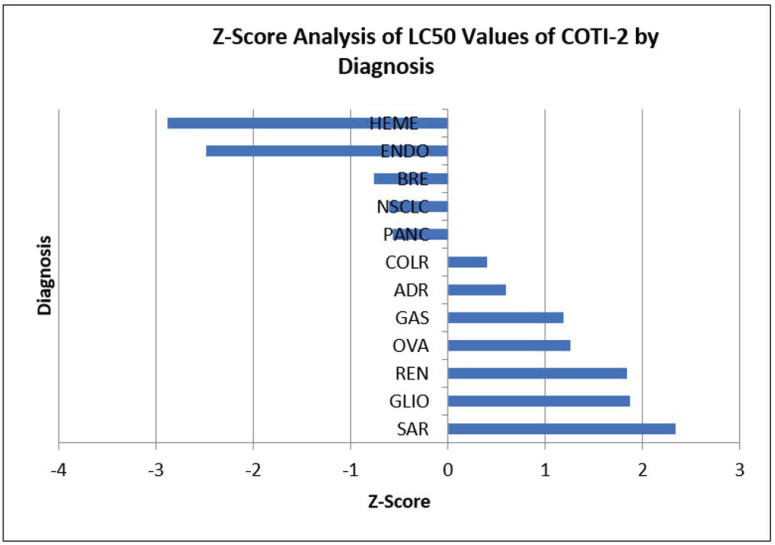
Lethal concentration (LC50) values were compared using Z-scores calculated by the formula Z = X − U/S, where X = Mean value, U = Sample value and S = Standard deviation. Results falling to the left of mean are more sensitive. Results falling to the right of mean are more resistant: Heme = Hematologic tumors; Endo = Endometrial; Adr = Adrenocortical; Colr = Colorectal; Panc = Pancreatic; SAR = Sarcoma; Ova = Ovarian; NSCLC = Non-small cell lung cancer; Gas = Gastric; Bre = Breast; Ren = Renal; Glio = Glioblastoma.

**Figure 3 genes-14-00747-f003:**
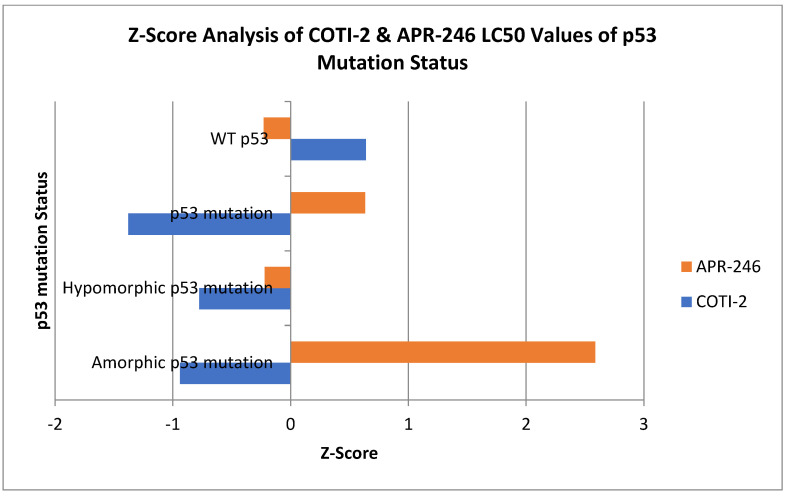
Z-Score analysis by P53 mutational status compares APR and COTI activity against wild-type versus mutated and amorphic (low or no functional protein) versus hypomorphic (altered protein structure) P53 subtypes. Results falling to the left of mean are more sensitive. Results falling to the right of mean are more resistant.

**Figure 4 genes-14-00747-f004:**
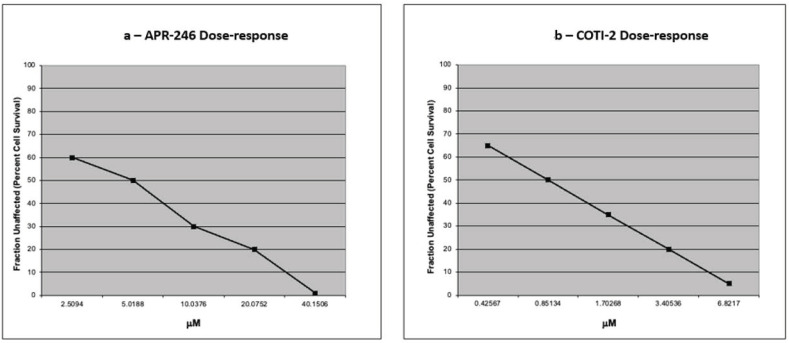
(**a**,**b**)—provide representative dose–response curves for APR-246 and COTI-2. Following 72 h drug exposure, tumor cell percent viability (0–100) measured against saline controls are provided in 5-point serial dilutions. Drug concentrations are provided as mMolar. Lethal concentration 50% (LC50) are interpolated using best line fit analyses.

**Table 1 genes-14-00747-t001:** Correlation Coefficients for COTI-2 and APR-246 Activity Against Other Drug Classes.

Drugs Correlated with COTI-2 and APR-246		R Value	*p* Value	Bonferroni Significance
APR-246	AZD6	0.005	0.978	No
	CISGEM	0.112	0.450	No
	DOX	0.413	0.015	No
	NM	0.173	0.344	No
	TAX	0.194	0.278	No
	CDDP	−0.258	0.213	No
	COTI-2	0.103	0.381	No
COTI-2	AZD6	0.321	0.00037	Yes
	CISGEM	0.330	0.000012	Yes
	DOX	0.347	0.000752	Yes
	NM	0.449	0.000019	Yes
	TAX	−0.036	0.726282	No
	CDDP	0.307	0.013542	No
	APR-246	0.103	0.381	No

Pearson Moment Correlation Coefficients for COTI-2 and APR-246 by Drug/Combination. Pearson Moment Correlations expressed as R values. Higher R values confer higher correlation. Significance calculated by Student T-test with α < 0.05 and Bonferroni calculation using the formula α/*n*, where *n* = number of comparisons performed. AZD6 = Selumetinib; CISGEM = cisplatin and gemcitabine; DOX = Doxorubicin; NM = Cytoxan; TAX = Taxol; CDDP = cisplatin.

## Data Availability

Data utilized in the preparation of this manuscript is reported in the manuscript and provided as Appendix A.

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
