# Peer review of "Therapeutic Targeting of P53: A Comparative Analysis of APR-246 and COTI-2 in Human Tumor Primary Culture 3-D Explants"

_genes, 2023, doi:10.3390/genes14030747_

Round 1

Reviewer 1 Report

In this paper authors attempted to investigate the therapeutic targeting of P53 using two novel compound APR-246 and COTI-2. The general idea is interesting and methods are appropriate for the journal requirement. However, paper needs minor revision in terms of language and content.

In the abstract, authors claimed that APR activity correlated with Doxorubicin 21 and Everolimus but I can not see such a trend in results section especially with Everolimus.

Introduction sections should be improved with more information related to APR-246 and COTI-2. The naming of these compounds should be consistent throughout the manuscript. In the current version, its inconsistent. 

Materials section does not have sufficient information. Authors could mention different statistical approaches used for various analysis. I am just curious how the Z-score was calculated. TThe authors' explanation for the Z-score in figure 1 (Z = X-U/S, where X = 131 Mean value, U = Sample value, and S = standard deviation) appears to be incorrect. 

How many mutations in TP53 gene were detected in this patient cohort? I would strongly suggest to add a table containing all TP53 mutations detected in these patients.

The full name of the drugs tested for correlation with APR-246 and COTI-2 should be mentioned in table. 1 legend. 

Author Response

Response to Reviewer 1 Comments

Point 1:

In the abstract, authors claimed that APR activity correlated with Doxorubicin 21 and Everolimus but I cannot see such a trend in results section especially with Everolimus.

Response 1:

We apologize for the oversight. The original abstract reflected an earlier and much smaller data set that suggested correlations between APR-246 and both Doxorubicin and Everolimus. With the larger data set (now 247 separate analyses) the correlation with Everolimus disappeared and the correlation with Doxorubicin markedly diminished. We appreciate the reviewers identifying our error and have corrected the manuscript to reflect our current and much larger data set.  See lines 20-22.

Point 2:

Introduction sections should be improved with more information related to APR-246 and COTI-2. The naming of these compounds should be consistent throughout the manuscript. In the current version, its inconsistent. 

Response 2:

We have added material on both APR-246 and COTI-2 to describe each compound and their modes of action and preliminary clinical experience with each. We standardized the naming of the compounds as APR-246 and COTI-2 throughout. For APR-246 we include the generic name Eprenetapopt. See lines 103-126.

Point 3:

Materials section does not have sufficient information. Authors could mention different statistical approaches used for various analysis. I am just curious how the Z-score was calculated. The authors' explanation for the Z-score in figure 1 (Z = X-U/S, where X = 131 Mean value, U = Sample value, and S = standard deviation) appears to be incorrect.

Response 3:

We have expanded the discussion of the Z-Score and statistical methods. See lines 191-232.

Point 4: 

How many mutations in TP53 gene were detected in this patient cohort? I would strongly suggest to add a table containing all TP53 mutations detected in these patients.

Response 4:

The specific TP53 mutations identified have now been included in supplemental Table 1.

Point 5:

The full name of the drugs tested for correlation with APR-246 and COTI-2 should be mentioned in table. 1 legend. 

Response 5:

APR-246 is Eprenetapopt. To our knowledge COTI-2 continues to be referred to as COTI-2.

Reviewer 2 Report

In the manuscript “Therapeutic targeting of P53: A comparative analysis of APR-246 and COTI-2 in human tumor primary culture 3-D explants” Nagourney and colleagues test activity of APR-246 and COTI in human tumor explants from 247 surgical specimens.

The introduction superficially discusses various aspects of p53 biology that are not always relevant to this study. At the same time, important points, such as the role of mutations in the p53 gene for different cancers and targeted therapies for cancers with wild-type and mutant p53, are not addressed.

Unfortunately, an adequate assessment of this manuscript is not possible for three reasons. First, the methodology and design of the experiment (details of the method used, controls, duration of treatment, drug concentrations, and so on) are not described. Second, sample data are missing (at least the number of samples for each cancer type/treatment and p53 genotypes for each sample must be present). Third, only generalized results are shown, from which it is impossible to judge the quality of the experiment and the adequacy of interpretation of the data obtained. The primary data, such as survival curves used for estimation of LC50, should be presented at least in the supplementary materials. It is recommended that the article be thoroughly revised.

Author Response

Response to Reviewer 2 Comments  

Point 1:

The introduction superficially discusses various aspects of p53 biology that are not always relevant to this study. At the same time, important points, such as the role of mutations in the p53 gene for different cancers and targeted therapies for cancers with wild-type and mutant p53, are not addressed.

Response 1:

We shortened the Introduction section (lines 28-72) and added text to describe the role of TP53 in different diseases to reflect the complexity of this gene’s role in tumor biology (lines 74-83).

Point 2:

Unfortunately, an adequate assessment of this manuscript is not possible for three reasons. First, the methodology and design of the experiment (details of the method used, controls, duration of treatment, drug concentrations, and so on) are not described.

Response 2:

We have expanded the Methods section to include a more granular description the methods and analyses applied. See lines 127-199). We have included a table of the identified TP53 mutations as Supplemental Table 1.

Point 3:

Third, only generalized results are shown, from which it is impossible to judge the quality of the experiment and the adequacy of interpretation of the data obtained. The primary data, such as survival curves used for estimation of LC50, should be presented at least in the supplementary materials. It is recommended that the article be thoroughly revised.

Response 3:

We have included representative dose response curves for both APR-246 and COTI-2 from individual tissue analyses. As the number of dose response curves in the data sets was large (in aggregate 247), we provide these examples to better describe the format and appearance of analyses. See Supplemental Figures S1 and S2. If the reviewer would like the data presented in a different format, we are happy to provide additional data.